# Hiccup-like Contractions in Mechanically Ventilated Patients: Individualized Treatment Guided by Transpulmonary Pressure

**DOI:** 10.3390/jpm13060984

**Published:** 2023-06-12

**Authors:** Evangelia Akoumianaki, Maria Bolaki, Georgios Prinianakis, Ioannis Konstantinou, Meropi Panagiotarakou, Katerina Vaporidi, Dimitrios Georgopoulos, Eumorfia Kondili

**Affiliations:** 1Department of Intensive Care Unit, University Hospital of Heraklion, 71110 Crete, Greece; maria_mpolaki@yahoo.gr (M.B.); prinian02@gmail.com (G.P.); konstantinou.ioannis@yahoo.gr (I.K.); mpanagiotarakou@gmail.com (M.P.); vaporidi@gmail.com (K.V.); kondylie@uoc.gr (E.K.); 2School of Medicine, University of Crete, 71003 Heraklion, Greece; georgopd@uoc.gr

**Keywords:** hiccups, respiratory myoclonus, esophageal pressure, patient-ventilator interaction, transpulmonary pressure, lung protective ventilation, VILI, multiple triggering, lung stress, strain, VIDD

## Abstract

Hiccups-like contractions, including hiccups, respiratory myoclonus, and diaphragmatic tremor, refer to involuntary, spasmodic, and inspiratory muscle contractions. They have been repeatedly described in mechanically ventilated patients, especially those with central nervous damage. Nevertheless, their effects on patient-ventilator interaction are largely unknown, and even more overlooked is their contribution to lung and diaphragm injury. We describe, for the first time, how the management of hiccup-like contractions was individualized based on esophageal and transpulmonary pressure measurements in three mechanically ventilated patients. The necessity or not of intervention was determined by the effects of these contractions on arterial blood gases, patient-ventilator synchrony, and lung stress. In addition, esophageal pressure permitted the titration of ventilator settings in a patient with hypoxemia and atelectasis secondary to hiccups and in whom sedatives failed to eliminate the contractions and muscle relaxants were contraindicated. This report highlights the importance of esophageal pressure monitoring in the clinical decision making of hiccup-like contractions in mechanically ventilated patients.

## 1. Introduction

Involuntary spasmodic inspiratory muscle contractions have been repeatedly reported in critically ill patients and can complicate the course of mechanical ventilation. Based on the frequency and characteristics of electromyographic discharge and the underlying pathophysiologic mechanism, these contractions can be classified as hiccups or as respiratory myoclonus and diaphragmatic tremor. The term ‘hiccup-like’ contractions is used to describe spasmodic muscle contractions resembling hiccups or respiratory myoclonus.

During hiccups, electromyography discloses the diaphragm and inspiratory intercostal muscles discharge, with sudden glottal closure that terminates inspiration and generates the characteristic “hic” sound [1,2,3]. A supraspinal mechanism is considered responsible for hiccups [4]. Respiratory myoclonus, also known as diaphragmatic flutter, diaphragmatic myoclonus, or Leeuwenhoek disease, refers to repetitive, involuntary contractions of the diaphragm or other respiratory muscles that have an average frequency of 150/min, irregular rhythm, and variable amplitude in electromyograph [5,6,7]. The pathophysiology includes abnormal excitation of the phrenic nerve due to central or peripheral disorders or irritation of the diaphragm itself. They usually cause respiratory distress and discomfort in the epigastrium in awake patients, although asymptomatic cases, termed “isolated diaphragmatic tremor”, have been described [5]. The differential diagnosis between respiratory myoclonus and hiccups requires electromyography of the diaphragm and/or other respiratory muscles. The management of hiccups or respiratory myoclonus aims to suppress muscle contractions with medicines (chloropromazine, haloperidol, olanzapine, benzodiazepines, muscle relaxants, metoclopramide, domperidone, etc.) or nonpharmacologic approaches (respiratory maneuvers, psychiatric therapy, phrenic nerve blockade) [1,8].

During invasive mechanical ventilation, the effects of hiccup-like contractions on gas exchange, potential injury to the lung and diaphragm, and patient-ventilator interaction are largely unknown. Furthermore it is unclear whether and how aggressively they should be treated. We report three mechanically ventilated patients with hiccup-like contractions who had esophageal pressure (Pes) monitoring for clinical reasons. One patient also had gastric pressure (Pgas) monitoring, which allowed the measurement of trandiaphragmatic pressure (Pdi). The patients were admitted in the mixed medico-surgical Intensive Care Unit of the University Hospital of Heraklion in Crete (Greece) between March and December of 2022. The effects of hiccup-like contractions on patient-ventilator interaction and lung stress were evaluated by measuring esophageal pressure and computing transpulmonary pressures. The transpulmonary pressure (PL) was calculated by substracting Pes from Paw (PL = Paw − Pes). Pdi was estimated as the difference between Pgas and Pes. Treatment was tailored according to these measurements.

## 2. Detailed Cases Description

### 2.1. Case 1

A 60-year-old man with a prior medical history of ischemic heart disease and dyslipidemia was admitted to the Intensive Care Unit (ICU) after decompressive craniectomy due to a large ischemic infarct of the right cerebellum. The patient was intubated, sedated and ventilated with volume-controlled mechanical ventilation (VCV). Antiepileptic therapy with levetiracetam 1 g/12 h was instituted and esophageal pressure monitoring was applied to facilitate lung protective ventilation. Five days after admission, the dose of sedatives was reduced, the patient regained spontaneous breathing activity and pressure support ventilation (PSV) with pressure support of 5 cmH_2_O, and positive end-expiratory pressure (PEEP) of 5 cmH_2_O was instituted. The Richmond Agitation and Sedation Scale was −3. However, further reduction of sedation was suspended due to multiple episodes of involuntary respiratory muscle contractions that were not associated with hypoxemia (PaO_2_/FiO_2_ 250–300 mmHg) but promoted significant patient-ventilator asynchrony in the form of multiple triggering, which persisted at all modes of assisted mechanical ventilation, caused respiratory alkalosis, and necessitated increased sedation and controlled MV. The episodes subsided transiently with bolus doses of lorazepam 2 mg and chloropromazine 25 mg, but recurred afterwards. Weaning attempts in the following three days failed because of these contractions. Inspection of airway flow, airway pressure (Paw), Pes, and Pdi over time demonstrated abrupt Pes deflections and Pdi inflections, with a frequency varying from 7 to 180/min. (Figure 1). Successive muscle contractions triggered the ventilator multiple times. During hiccup-like contractions, swings in Pes (ΔPes), Pdi (ΔPdi), and PL (ΔPL) as high as 22 cmH_2_O, 20 cmH_2_O, and 26 cmH_2_O, respectively, were noted, along with extremely high tidal volumes (VT) of up to 16 mL/kg PBW. These VTs were almost two times higher than those inflated during hiccup-free breathing. They resulted from either large Pes swings and/or multiple ventilator triggering. Hiccup-like contractions persisted under sedation with propofol (280 mg/h, 12 mg/kg), remifentanyl 700 μg/h, and midazolam 5 mg/h, and necessitated muscle relaxation with cis-atracurium as a continuous infusion of 10 mg/h for 4 days. Finally, chloropromazine (25 mg/8 h) in addition to levetiracetam (1 g/12 h) eliminated spasmodic diaphragmatic contractions and sedation and muscle relaxants were withdrawn on day 15 in ICU. The patient regained consciousness and exited the ICU 21 days after his admission with a GCS of 14/15.

### 2.2. Case 2

A 58-year-old man was admitted to the ICU after surgical repair of an abdominal aortic aneurysm. He had a medical history of cardiovascular diseases, including coronary artery disease with previous myocardial infarction treated with angioplasty, left ventricular systolic dysfunction with left ventricular ejection fraction of 30%, end-stage renal disease necessitating hemodialysis 3 times per week, chronic obstructive pulmonary disease, and atrial fibrillation. During the operation, he developed severe bleeding treated with a massive transfusion of blood products. Upon ICU admission, the patient was sedated and he was receiving high doses of noradrenaline (0.5 μg/kg/min) and vasopressin (0.03 IU/h) due to hemodynamic instability. The patient was managed with intravenous fluids and vasopressors and was sedated with intravenous propofol (280 mg/h) and remifentanyl (700 μg/h). Fluid resuscitation resulted in significant improvement of hemodynamic instability and metabolic acidosis in the next 48 h. Nevertheless, hypoxemia progressively worsened with a PaO_2_/FiO_2_ between 100–130 mmHg and neuromuscular blockade with cis-atracurium was added to sedatives. On ICU day 4, muscle relaxation was withdrawn because of acute CPK elevation (3500 U/lt). At that time, the patient was ventilated with VCV, with VT of 6 mL/kg PBW, had a respiratory rate of 27 br/min, FiO_2_ of 55%, and PEEP of 18 cmH_2_O. The PaO_2_/FiO_2_ was 146 mmHg. A few hours following cis-atracurium discontinuance, he developed hiccup-like contractions, with a mean frequency of 22 contractions per min. There were no signs of chronic obstructive pulmonary disease exacerbation on lung auscultation or ventilator waveforms inspection. No breath stacking or other forms of major asynchrony were noted on the ventilator screen, but hypoxemia worsened following the emergence of contractions, with a PaO_2_/FiO_2_ of 100 mmHg. The level of PEEP increased to 20 cmH_2_O, but hypoxemia did not improve, and an esophageal catheter was introduced to titrate the ventilator settings. Swings in Pes (ΔPes) and PL (ΔPL) resulting from hiccup-like contractions were relatively low: 7 cmH_2_O and 6–7 cmH_2_O, respectively. Notwithstanding this, the end-expiratory PL was negative, indicating atelectasis (Figure 2A). The contractions did not initially respond to metoclopramide, baclofen, and intravenous midazolam of 15 mg/h. PEEP increased to 25 cmH_2_O to prevent negative end-expiratory PL. Hypoxemia significantly improved following PEEP escalation (PaO_2_/FiO_2_ rose to 167 mmHg). The end-expiratory PL was 5 cmH_2_O. Despite very high PEEP levels, the driving PL was 6 cmH_2_O, well below the upper thresholds associated with lung injury (Figure 2B). End-inspiratory PL, calculated by subtracting end-inspiratory Pes from P_plateau_, was 11 cmH_2_O. Maintenance of high PEEP levels along with regular measurements of Pes and PL during end-inspiratory and end-expiratory pauses resulted in further PaO_2_/FiO_2_ improvement over the next 24 h, although hiccups persisted. In the following days, metoclopramide and baclofen eliminated hiccups and PEEP was gradually reduced.

### 2.3. Case 3

A 73 year-old-man with a medical history of multiple myeloma and chronic obstructive pulmonary disease was hospitalized for meningitis from Streptococcus pneumonia. He was treated with intravenous ceftriaxone 2 g/12 h and dexamethasone, but three days after hospital admission his level of consciousness deteriorated and he was transferred to the ICU where he was intubated with a Glascow Coma Scale of 7/15. On ICU day 2, he developed seizures and midazolam (10 mg/h iv) and levetiracetam (1 g/12 h iv) were instituted. An esophageal catheter was inserted to facilitate lung protective ventilation during controlled MV. Over the next days, the patient’s clinical condition gradually improved. Intravenous sedation was withdrawn 7 days after ICU admission and PSV was applied with an assist level of 5 cmH_2_O and PEEP of 8 cmH_2_O. Upon sedation discontinuation, hiccup-like contractions were noticed on clinical examination. Similar to the first case, abrupt Pes deflections were documented, with a frequency varying between 5 to 190/min, indicating possible respiratory myoclonus. The maximum ΔPes and ΔPL of these contractions were 15 cmH_2_O, and 19 cmH_2_O, respectively, and triggered the ventilator multiple times. Multiple triggering, however, neither increased VT nor caused ABGs derangement (Figure 3). An increase in the dose of levetiracetam to 1 g/8 h and an addition of lacosamide (150 mg/12 h) stopped hiccup-like contractions. No sedatives were required and the patient remained on assisted ventilation. He was weaned from the ventilator 19 days after intubation with complete recovery of brain function and no relapse of respiratory myoclonus.

## 3. Discussion

To our knowledge, this is the first analysis of the effects of hiccup-like contractions on lung stress and strain and patient-ventilator interaction in mechanically ventilated patients. To examine these effects, we measured esophageal and transpulmonary pressure changes during bursts of spasmodic diaphragmatic contractions. In one patient, gastric pressure monitoring was available and transdiaphragmatic pressure was also evaluated. We used the term ‘hiccup-like’ events as we could not differentiate whether these spasmodic diaphragmatic contractions resembled hiccups or respiratory myoclonus. Such a distinction required electromyography, which was beyond the scope of our investigation. It is uncertain whether the first and third patient (Case 1 and 3) in this report suffered from respiratory myoclonus, diaphragm tremor, or hiccups. Zero flow due to glottic closure is a cardinal characteristic of hiccups in awake patients with spontaneous breathing. Nevertheless, it does not occur in critically ill patients because of endotracheal intubation. Esophageal pressure monitoring disclosed that hiccup-like contractions had different effects in the respiratory system of the patients. In two patients, these contractions caused significant patient-ventilator asynchrony in the form of multiple triggering. In addition, they were associated with extremely high tidal changes in esophageal and transpulmonary pressures and very high transdiaphragmatic pressures in the patient who had gastric pressure monitoring. In the other patient, hiccups promoted progressive atelectasis as indicated by the negative end-expiratory PL. These different effects necessitated different management that was individualized based on Pes and PL measurements.

### 3.1. Hiccups-like Contractions and Dynamic Lung Stress and Strain

Resumption of spontaneous breathing is a double-edged sword, with both positive and negative consequences in patients with injured lungs. Inspiratory muscle efforts reduce pleural pressure, increase transpulmonary pressure, and inflate the lung. Under passive MV, the cephalic displacement of the diaphragm favors ventilation of the nondependent alveolar units. During spontaneous breaths, the aeration of dependent lung units is enhanced due to diaphragm movement, ventilation can become more homogenous, and gas exchange can be improved [9]. Furthermore, inspiratory efforts redistribute pulmonary blood flow toward the nondependent parts of the lung [10,11]. On the other hand, excessive inspiratory muscle contraction can result in very negative transpulmonary pressures that can move intravascular fluid to interstitial tissue and alveoli, resulting in pulmonary edema [12,13,14]. Additionally, they promote high transpulmonary pressure and large tidal volume, both linked to lung damage. Animal studies have shown that abrupt deflation, following a sustained lung inflation, could induce acute lung injury mediated by acute left ventricular decompensation as a result of increased left ventricular preload and afterload [11,15]. Finally, diaphragmatic contractions may elicit lung injury irrespective of VT changes, secondary to a phenomenon known as ‘Pendelluft’. Pendelluft refers to the movement of gas from the nondependent to dependent lung regions as a consequence of uneven distribution of the negative pressure generated by the diaphragmatic contraction during assisted ventilation. During inspiratory muscle contractions, dependent lung units are early inflated with concurrent deflation of nondependent alveoli. Consequently, pendelluft causes regional overdistention in dependent regions of already injured lungs, even at low tidal volumes [11]. The higher the inhomogeneity of the lung, the greater the overdistention of dependent alveoli at the same magnitude of inspiratory effort [16,17]. Furthermore, recent studies have shown that the higher the intensity of diaphragmatic contraction, as measured via the electrical activity of the diaphragm (Edimax) or Pes, the higher the impact of pendelluft in lung overstretching [18,19]. Notwithstanding, the relationship between the magnitude of inspiratory effort and pendelluft is not as straightforward as recent data suggest [18,20]. The frequency of high magnitude pendelluft might be also linked to the severity of inflammatory response following inspiratory efforts in ARDS patients [20]. 

The preponderance of benefits over harmful effects on the lung depends on the severity of the lung injury and on the magnitude of inspiratory muscle contraction [17]. Monitoring of Pes in our patients permitted the evaluation of the magnitude of inspiratory muscle contraction. Hiccup-like contractions were associated with very high ΔPes in the first patient and the third patient—22 and 15 cmH_2_O, respectively. Inflated VTs were excessive—up to 16 mL/kg PBW in the first case. The influence of such intense contractions on pendelluft development and its magnitude could not be assessed without electrical impedance tomography but the theoretical risk is considerable. Moreover, significant increases to vascular transmural pressure secondary to large ΔPes can be assumed. Static ΔPLs could not be measured because of hiccups but dynamic ΔPL, the difference between minimum and maximum PL value during a ventilator assisted breath, was assessed. Dynamic PL resembles dynamic lung stress during inflation and, although protective values are largely unknown, animal studies have indicated the development of lung injury at values higher than 15–20 cmH_2_O [21]. In both patients, spasmodic contractions induced potentially injurious dynamic ΔPLs. In the first patient, dynamic ΔPL during these contractions was as high as 26 cmH_2_O, and these high values were observed several times over the recording period. Notably, high ΔPL was observed at only 5 cmH_2_O level of support. There is a linear relationship between lung stress strain as described by the equation: ΔPL (stress) = specific elastance × strain. Considering a value of specific elastance of 13.5 cmH_2_O, lung strain in the first and third patient were 1.93 and 1.41, respectively [22]. A harmful threshold of strain is considered at around 1.5. Therefore, the first patient had values exceeding this threshold whilst strain of the third patient was very close to the upper safe value. Synchronous addition of higher pressure support level to an already highly negative pleural pressure would further augment ΔPL because hiccup-like contractions are not related to drive and ΔPes would remain unaffected from ventilator assistance. 

### 3.2. Hiccup-like Contractions and Static Lung Stress

To eliminate the risk of ventilator-induced lung injury (VILI), the literature suggests maintaining static ΔPL < 10–12 cmH_2_O and end-inspiratory PL < 20–25 cmH_2_O. Nevertheless, apart from end-inspiratory PL, end-expiratory PL is also important since negative values imply alveolar collapse at end-expiration, another important mechanism of lung injury. Alveolar collapse worsens oxygenation, reduces lung compliance, increases lung inhomogeneity, and, when accompanied by alveolar opening and closure during inspiration and expiration, can cause atelectrauma. To eliminate alveolar collapse, PEEP should be titrated to target a positive end-expiratory transpulmonary pressure [9,23,24]. There are two methods to estimate PL from Pes: the direct method and the elastance-derived method. The direct method calculates PL by subtracting Pes from Palv during end-inspiratory and end-expiratory occlusions. The elastance-derived method assumes that PL equals the product of Paw times the ratio of lung elastance (E_L_) to respiratory system elastance (E_Rs_): PL = Paw × E_L_/E_Rs_. There is considerable controversy as to which method better estimates transpulmonary pressures [25]. Recent experiments demonstrated that the direct method more accurately reflects PL in the mid-dorsal and dependent regions, but may underestimate PL and, thus, lung strain and overdistention in the nondependent lung units. The reason is that Pes might overestimate pleural pressure in the nondependent parts of the lung [26]. Thus, the direct method seems preferable to titrate PEEP while the elastance derived method is calculated to end-inspiratory stress.

In the second patient of our report, hiccup-like contractions resulted in progressive hypoxemia. Sedatives were given at maximum doses and muscle relaxants were contraindicated due to CPK elevation. Considerably high PEEP levels failed to improve oxygenation and carried the risk of lung overdistention without knowing the resulting end-inspiratory PL and ΔPL. The introduction of esophageal catheter suggested that the mechanism of hypoxemia was atelectasis, as denoted by the negative end-expiratory PL. Further PEEP increase to target a positive end-expiratory PL succeeded to improve hypoxemia. Concurrently, Pes measurement ensured that static ΔPL at very high PEEP levels (25 cmH_2_O) was only 6 cmH_2_O, well below the upper safe limit of 12 cmH_2_O. End-inspiratory PL calculated by the direct method was only 11 cmH_2_O. According to the elastance-derived method, end-inspiratory PL was 22 cmH_2_O, just above the upper safe limit. The reanalysis of the EPVent-1 data demonstrated that mortality was associated with a low ΔPL [26]. Based on the very low ΔPL measured in our patient along with better oxygenation and less atelectasis, high PEEP levels were considered safer than lower PEEP levels. 

### 3.3. Hiccup-like Contractions and Patient-Ventilator Synchrony

Except from injurious PL, bursts of hiccup-like contractions caused multiple ventilator triggering in two patients. Multiple ventilator triggering may expose patients to high VT, as happened in the first patient, resulting in significant lung stress and potentially harmful strain, and they are associated with erroneous ventilatory rate, high minute ventilation, and respiratory alkalosis [27,28]. Moreover, clusters of multiple triggering, such as those observed during bursts of hiccup-like contractions, have been recently associated with worse clinical outcomes in ICU patients [29]. Similar to our observations, Menga et al. used recent electrical activity of the diaphragm to demonstrate that diaphragmatic myoclonus in a critically ill patient promoted patient-ventilator asynchronies and high VT [30]. Nevertheless, the intensity of diaphragmatic contractions or associated PL were not measured in that study. 

### 3.4. Hiccup-like Contractions and Diaphragm

Maintenance of diaphragmatic activity mitigates the risk of diaphragmatic atrophy and prolonged weaning [31,32]. The incidence of ventilator-induced diaphragmatic dysfunction (VIDD) in mechanically ventilated patients ranges between 23–84% and diaphragm inactivity has been strongly correlated with its development [31,33]. On the other hand, excessive diaphragmatic contraction may also result in deleterious anatomical and functional modifications of the diaphragm [31], contributing to VIDD and prolonged MV [23,34,35,36,37,38,39,40]. To balance between diaphragmatic inactivity and overuse, the quantification of the inspiratory effort through Pes and Pdi measurement has been proposed. A ΔPes higher than 8–12 cmH_2_O and ΔPdi above 15 cmH_2_O are considered as markers of diaphragmatic overuse, although further studies are required to confirm that intense inspiratory contractions cause diaphragmatic damage [23,41]. In two patients, Pes swings higher than 15 cmH_2_O resembled strenuous inspiratory muscle contractions [42]. It is unknown whether involuntary intense diaphragmatic contractions impact similarly the diaphragm as diaphragmatic contractions secondary to high respiratory drive [42]. Additionally, hiccup-like contractions are not synchronous to mechanical breath, subjecting the diaphragm to the risk of isometric or eccentric contractions. Experimental data suggest that eccentric diaphragmatic contractions may injure muscle fibers and this mechanism could explain why patient-ventilator asynchronies have been related to worse outcomes in mechanically ventilated patients [43,44,45,46].

### 3.5. Unanswered Questions

Although our observations indicate that hiccup-like contractions may have various injurious effects in the respiratory system of mechanically ventilated patients and that esophageal pressure monitoring might help to identify and address them, there are several questions that need to be answered. The exact incidence of such contractions in mechanically ventilated patients and their association with specific patients groups needs to be explored. Additionally, how often hiccup-like contractions lead to lung and/or diaphragm injury deserves further investigation. The quantification of pendelluft requires electrical impedance tomography, which was not applied in our cases. Based on only three cases, no recommendations can be made on how often we need to monitor Pes and PL and what actions should be taken in case of these involuntary contractions. These cases encourage, however, a more thorough investigation of the effects of hiccup-like contractions on the lungs and diaphragm.

## 4. Conclusions

We demonstrate, for the first time, that significant patient-ventilator asynchrony and injurious pressures can be exerted on the lung and the diaphragm during contractions secondary to hiccups or respiratory myoclonus. The effects of these contractions on the respiratory system vary and the necessity of intervention as well as the type of management should be individualized. Esophageal pressure monitoring may aid clinicians to more accurately estimate the risk of these contractions and adjust interventions accordingly. This is especially relevant in patients at risk of lung or diaphragm injury.

## Figures and Tables

**Figure 1 jpm-13-00984-f001:**
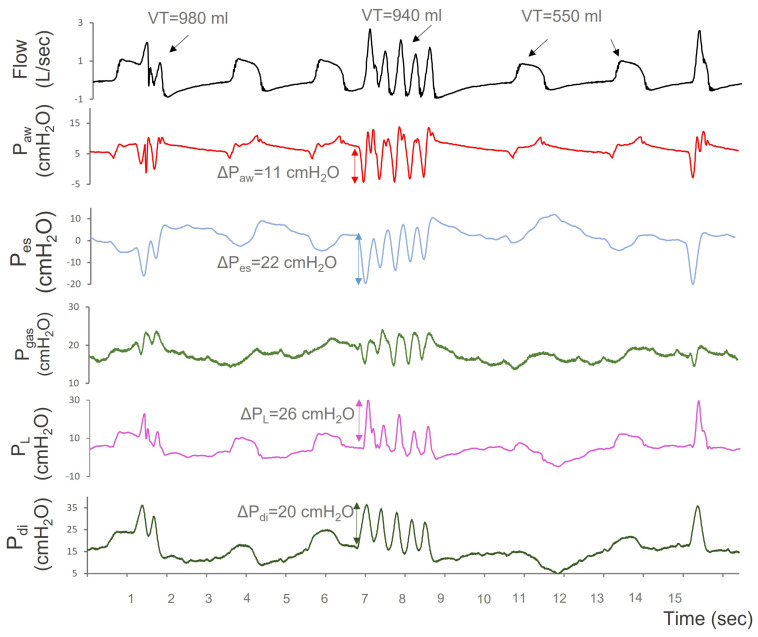
Flow, airway pressure (Paw), esophageal pressure (Pes), gastric pressure (Pgas), transpulmonary pressure (PL), and transdiaphragmatic pressure (Pdi) over time (s) during episodes of hiccups. P_L_ was calculated by subtracting Pes from Paw. The patient is ventilated with pressure support ventilation with a level of support of 5 cmH_2_O and positive end-expiratory pressure of 6 cmH_2_O. During hiccup-like episodes, swings in Pes (blue arrow) and PL (purple arrow) were ΔPes = 22 cmH_2_O and ΔPL = 26 cmH_2_O, respectively, resulting in high flows and tidal volumes. Pdi swings were also very high (20 cmH_2_O), indicating intense diaphragmatic contractions. After the third ventilator breath, a burst of five inspiratory muscle contractions can be noticed, resulting in VT of 940 mL. Similar bursts of hiccup-like contractions were seen several times over the recording period.

**Figure 2 jpm-13-00984-f002:**
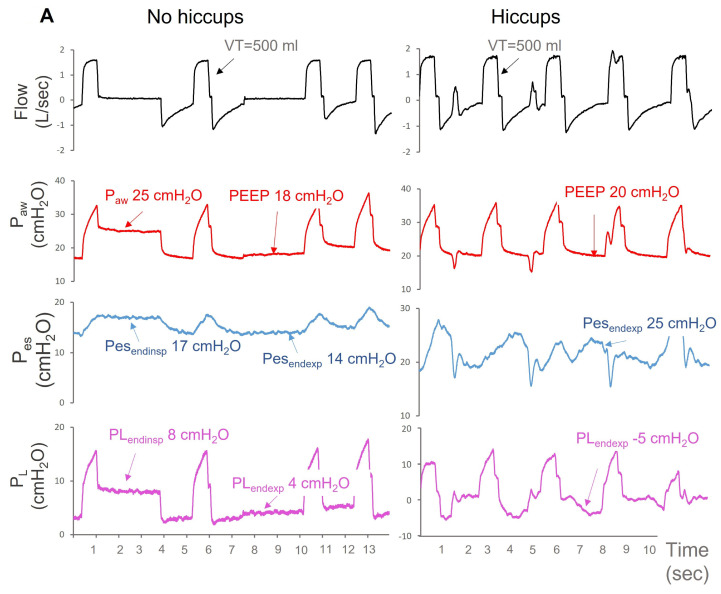
Flow, airway pressure (Paw), esophageal pressure (Pes), and transpulmonary pressure (PL) over time (s) in a patient ventilated with volume controlled ventilation (VCV) with a VT of 500 mL (6 mL/kg PBW), positive end-expiratory pressure (PEEP) of 18 cmH_2_O, and respiratory rate of 27 breaths/min. PL was calculated by subtracting Pes from Paw (**A**). Before hiccup-like contractions (left panel of (**A**)), end-inspiratory and end-expiratory pauses (arrows) disclosed end-inspiratory (PL_endinsp_) and end-expiratory PL (PL_endexp_) of 8 and 4 cmH_2_O (purple arrows), respectively. Following the emergence of hiccup-like contractions emergence (right panel of (**A**)), hypoxemia led to PEEP increase at 20 cmH_2_O. Despite PEEP increase, PL_endexp_ (purple arrows) was negative and hypoxemia worsened. When PEEP increased to 25 cmH_2_O, a positive PL_endexp_ was sustained and hypoxemia improved (**B**). Despite high PEEP levels and high Paw plateau levels (34 cmH_2_O), PL_endinsp_ calculated with the direct method (subtraction of Pes from Paw) was well below the upper safe threshold of 20 cmH_2_O and driving PL was also low (6 cmH_2_O). The elastance-derived calculation of PL_endinsp_ disclosed a value of 22 cmH_2_O.

**Figure 3 jpm-13-00984-f003:**
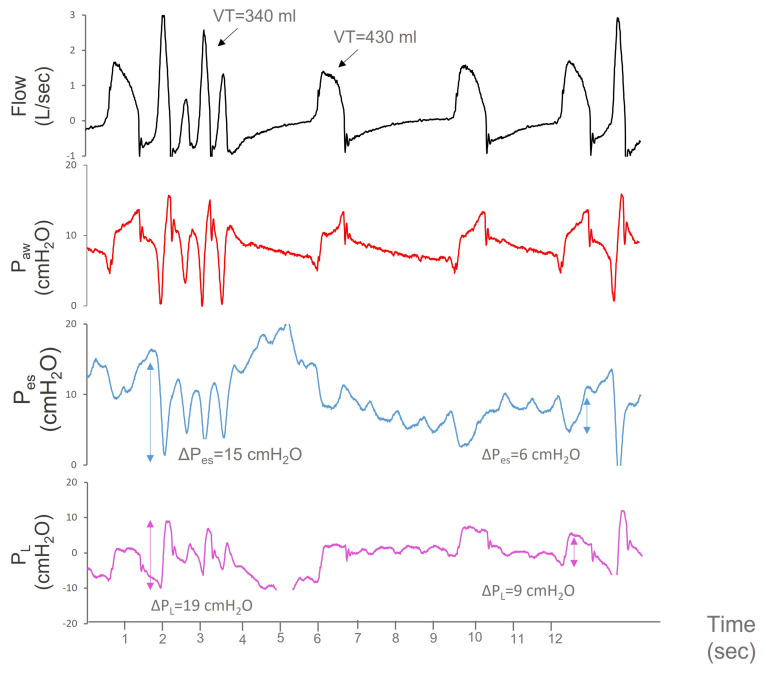
Flow, airway pressure (Paw), esophageal pressure (Pes), and transpulmonary pressure (PL) over time (s) during a burst of hiccup-like contractions. PL was calculated by subtracting Pes from Paw. The patient is ventilated with pressure support ventilation with a level of support of 5 cmH_2_O and positive end-expiratory pressure of 8 cmH_2_O. During normal breaths, the average ΔPes and ΔPL are 6 cmH_2_O and 9 cmH_2_O, respectively, and are associated with tidal volumes (VT) of ≈430 mL (6 mL/kg PBW). During hiccup-like episodes, large swings in Pes (blue arrow, ΔPes = 15 cmH_2_O) and PL (purple arrow, ΔPL = 19 cmH_2_O) can be noted. Bursts of hiccup-like contractions promoted multiple ventilator triggering. Multiple triggering was not associated with high VT (340 mL, 5 mL/kg PBW).

## Data Availability

All data included in this study can be provided by the corresponding author upon reasonable request.

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
