# Peer review of "Hiccup-like Contractions in Mechanically Ventilated Patients: Individualized Treatment Guided by Transpulmonary Pressure"

_jpm, 2023, doi:10.3390/jpm13060984_

Round 1

Reviewer 1 Report

The manuscript depicted 3 cases of hiccups-like contraction in mechanically ventilated patients treated under transpulmonary pressure monitoring.

The authors presented in case report form and emphasized on the potential risk of hiccups in ventilated patients from a physiological point of view.

The authors are respiratory physiologists and emphasized the importance of stress and strain and end-expiratory transpulmonary pressure. This manuscript is instructive because there is no article dealing with PL in ventilated patients with hiccup. The weak points are the lack of gastric pressure and EMG measurement. the term hiccups-like contractions were first mentioned by the authors. The viewpoints raised by the authors are just current consensus. The actual effect of hiccup in ventilated patients remains to be investigated.

My comment:

Introduction:

1.      Line 72: type error: Ritchmond -Richmond.

2.      Case 1: Figure 1: DPes change to ∆Pes.

3.      Case 2 : Figure 2A: The swings in Pes is small (7 cmH2O) and regular. Usually for hiccup, the peak to trough duration of ∆Pes is less than 0.2 second. So the hiccup-like breaths are reliable. As gastric pressure was not measured, the negative end-expiratory PL in hiccup-like breaths could be either inspiratory muscle relaxation or expiratory muscle contraction. For end-inspiratory PL, I think the elastance ratio method may be more appropriate (the author could add) although some difference existed with patient under active breath.

4.      line 148: Case 2: figure legend: (PLendinsp) and end-expiratory PL (PLendexp) of 14 and 4 cmH2O—14 should be 8.

5.      Case 3: Figure 3: DPes and DPL change to ∆Pes and ∆PL.

6.      Figure 3 legend: some cmH2O—change to cmH2O

7.      Line 220 to 222: Specific elastance actually varied with each patient and derived from static PL. Case 3 is a case of COPD, therefore the use of dynamic PL to represent lung stress should be cautious.

8.      Line 228 to 230: ARDS lung with heterogeneous distribution where focal solid areas may resist shape deformation and thus may cause imperfect elastic anisotropic inflation and facilitate pendelluft. Therefore, pendelluft may not be present in cases without ARDS as the non-injured lung may be more homogeneous and less likely to have pendelluft, the author may take this into consideration.

9.      Line 316: Refernce 4: absence of Journal name

Author Response

We sincerely thank the editor and reviewers for their constructive comments. In the revised version all point raised were addressed and we think that contributed to significant improvements compared to the initial manuscript. The new version is longer, close to 4000 words and more updated references have been added. Except from discussing in more detail the mechanisms and measurements of lung and diaphragmatic injury, we changed all Figures in the article in an effort to make them more comprehensive and easy to understand.

Detailed response to reviewer 1

  1. Line 72: type error: Ritchmond -Richmond.

Answer: The reviewer is right. The type error has been corrected.

  1. Case 1: Figure 1: DPes change to ∆Pes.

Answer: The change is included in the revised version

  1. Case 2 : Figure 2A: The swings in Pes is small (7 cmH2O) and regular. Usually for hiccup, the peak to trough duration of ∆Pes is less than 0.2 second. So the hiccup-like breaths are reliable. As gastric pressure was not measured, the negative end-expiratory PL in hiccup-like breaths could be either inspiratory muscle relaxation or expiratory muscle contraction. For end-inspiratory PL, I think the elastance ratio method may be more appropriate (the author could add) although some difference existed with patient under active breath.

Answer: Gastric pressure was indeed not measured as the reviewer correctly points out. Nevertheless, a negative end-expiratory PL, even if expiratory muscle induced, indicates atelectasis. End-inspiratory PL calculated with the elastance-derived method is now included both in the text and in the figure legend. 

  1. line 148: Case 2: figure legend: (PLendinsp) and end-expiratory PL (PLendexp) of 14 and 4 cmH2O—14 should be 8.

Thank you for your comment. There was indeed an error in calculation and it is now corrected.

  1. Case 3: Figure 3: DPes and DPL change to ∆Pes and ∆PL.

Answer: changes were made according to reviewers suggestions.

  1. Figure 3 legend: some cmH2O—change to cmH2O

Answer: changes were made in line with your remark.

  1. Line 220 to 222: Specific elastance actually varied with each patient and derived from static PL. Case 3 is a case of COPD, therefore the use of dynamic PL to represent lung stress should be cautious.

Answer: The reviewer is right, if resistance is high, dynamic ΔPL incorporates resistance. Nevertheless, the patient was not ventilated due to COPD exacerbation and flow-time waveforms as well as resistance measurements when he was sedated did not indicate resistive pattern. Hence, in this particular case, contribution of resistance to dynamic PL is similar to the usual, non-obstructive critically ill patient.

  1. Line 228 to 230: ARDS lung with heterogeneous distribution where focal solid areas may resist shape deformation and thus may cause imperfect elastic anisotropic inflation and facilitate pendelluft. Therefore, pendelluft may not be present in cases without ARDS as the non-injured lung may be more homogeneous and less likely to have pendelluft, the author may take this into consideration

Answer: These are very relevant comments and we thank the reviewer for them. In the new version, a paragraph discussing the factors affecting pendelluft, including that its significance depends on the severity of lung injury, is included.

  1. Line 316: Reference 4: absence of Journal name

Answer: the reference is no correct.

Author Response

We sincerely thank the editor and reviewers for their constructive comments. In the revised version all point raised were addressed and we think that contributed to significant improvements compared to the initial manuscript. The new version is longer, close to 4000 words and more updated references have been added. Except from discussing in more detail the mechanisms and measurements of lung and diaphragmatic injury, we changed all Figures in the article in an effort to make them more comprehensive and easy to understand.

Detailed response to reviewer 2

The sentences: ..However, dynamic ΔPL, the difference between minimum and maximum PL value during a ventilator assisted breath, 209 could be assessed. Dynamic PL resembles maximum stress during inflation and, although protective values are largely unknown, animal studies have indicated the development of lung injury at values higher than 15-20 cmH2O12“ is for the reader unclear – are you discusSing dynamic driving transpulmonary pressure or dynamic inspirátory transpulmonary pressure (I miss delta sign in the second sentence, line 210).

Answer: Thank your for this constructive comment. The sentences have been rephrased and new references as well as clarifications regarding transpulmonary pressure have been adde. In addition, we separate the discussion in two new paragraphs, discussing separately static and dynamic stress.

Transpulmonary pressure was calculated as a direct subtraction or Paw – Pes. This approach has some limitation to correctly estimate inspiratory transpulmonary pressure in the non-dependent lung regions. The limit of inspiratory PL suggested by Mauri is appropriate when inspirátory PL is calculated using elastance-derived method.

Answer: We totally agree with this comment. Now, end-inspiratory PL is calculated also with the elastance derived method and a more detailed discussion about PL calculation with the two methods has been added.

Controversy exists, if the suggested range of inspirátory transpulmonary could be considered safe -please include this topis into discussion. Units follow numbers sometimes with and sometimes without space H20 line 84 /zero instead of O) Grey area at Figure 1 does not follow the flow -time curve and si extended below the zero line DPes and DPL is used instead of ΔPes and ΔPL in fugure legend, which makes the picture more difficult to understand.

Answer: We have included a discussion about the safe range of inspiratory PL. We also corrected units adding space between them and the numbers throughout the text. Grey area has been removed from the Figures